# Plant-Derived Immunomodulatory Nanoadjuvants for Cancer Vaccines: Current Status and Future Opportunities

**DOI:** 10.3390/vaccines13040378

**Published:** 2025-03-31

**Authors:** Yimin Jia, Hui Zhu, Xinyu Cai, Cun Sun, Yan Ye, Dingyi Cai, Shuaifei Yang, Jingjing Cheng, Jining Gao, Yun Yang, Hao Zeng, Quanming Zou, Jieping Li, Hongwu Sun, Wenxiu Wang

**Affiliations:** 1Chongqing University Cancer Hospital, Chongqing 400030, China; jiayimin@cqu.edu.cn; 2Department of Microbiology and Biochemical Pharmacy, College of Pharmacy, Third Military Medical University, Chongqing 400038, China; 19122490822@163.com (H.Z.); cxinyu0127@163.com (X.C.); suncun_yl@163.com (C.S.); yeyan1028@163.com (Y.Y.); cdy2000163@163.com (D.C.); 18184761952@163.com (S.Y.); 17790304889@163.com (J.C.); jngao@163.com (J.G.); yangyun@tmmu.edu.cn (Y.Y.); zeng1109@163.com (H.Z.); qmzou2007@163.com (Q.Z.); 3Department of Stomatology, The 79th Group Army Hospital of PLA, Liaoyang 111000, China; 4Affiliated Nanhua Hospital of University of South China, Hengyang 421002, China; xqdoctor_li@163.com; 5Shandong Binzhou Animal Science and Veterinary Medicine Academy, Binzhou 256600, China; 6Shandong Academician Workstation, Binzhou 256600, China

**Keywords:** plant-derived adjuvants, cancer vaccine, polysaccharides, saponin, immune response

## Abstract

Cancer is a major cause of death worldwide, and vaccine administration is an effective way to stimulate immune responses in patients and to achieve preventive and therapeutic effects. Few vaccines have been used in clinical settings because they have poor immunogenicity, and it is difficult to induce a robust immune response in patients. An adjuvant is an important component of a vaccine that can enhance the intensity, speed, and duration of immune responses. The achievements of adjuvants in the production of stable, safe, and immunogenic tumor vaccines have aroused the enthusiasm of researchers. Recent results have suggested that plant-derived adjuvants have unique advantages, such as greatly improving immune responses to cancer vaccines and promoting humoral and cellular immunity with good biocompatibility and biodegradability. When these adjuvants are used in combination with vaccines, they can not only activate the immune response in vivo but can also promote cytokine secretion and accelerate dendritic cell maturation. This review focused on the application progress of plant adjuvants, including saponins, polysaccharides, flavonoids, and plant virus-like particles, and their combination with nano-delivery systems in cancer vaccines. At the same time, we have also discussed the immunomodulatory mechanisms of these adjuvants and their prospects for improving vaccine efficacy in the treatment of cancer in the future. These promising plant adjuvants may provide prospects and a research basis for the development of tumor vaccines.

## 1. Introduction 

In recent years, cancer has become the leading cause of death worldwide, resulting in an estimated 10 million deaths globally in 2020 [1], and in 2023 it was estimated to result in about 1,958,310 new cancer cases and 609,820 cancer deaths in the United States. While most patients undergo standard treatments including targeted therapy, chemotherapy, and radiation, the cancer still relapses. However, the development of vaccine treatment has attracted great attention because of its high specificity and lower toxicity. Previous data have demonstrated high vaccine efficacy in murine tumor models. Furthermore, there have also been promising research results in early-stage human clinical trials for prostate cancer, lung cancer, and breast cancer. However, clinical responses to these vaccines and the clinical efficacy have not been satisfactory in phase III trials. This is because these vaccines, which do not use an effective adjuvant or delivery system, are unable to induce robust CD8^+^ T immune responses and generally lack the triggering of B cell-mediated humoral immunity [2]. For example, for patients diagnosed with human papillomavirus infection (HPV)-associated cancers, their immune systems tolerate cancer cells or are suppressed due to the disturbance of the antigen presentation process and the activation of CD8^+^ and CD4^+^ T cells. Therefore, vaccines, with the help of adjuvants, can significantly improve HPV-specific T-cell immune responses, kill the cancer cells, and exert immune protective effects [3,4].

Ideal cancer vaccines would be effective in stimulating immune responses [5]. Generally, protein antigens are weakly immunogenic, and adjuvants are essential to trigger strong and prolonged activation of the immune system. Adjuvants have the advantages of overcoming immune tolerance and maximizing immune responses because they can play a key role in promoting immune responses through the following pathways: (1) maintaining release locally at the injection site, resulting in the “depot” effect; (2) recruiting immune cells to create an immune-competent environment; (3) recognizing and activating recruited antigen-presenting cells (APCs) to promote DC maturation, which increases the expression of MHC I (major histocompatibility complex I) or MHC II (major histocompatibility complex II) and further activates costimulatory signals such as CD40 and CD80/86; and (4) increasing the antigen migration of these activated APCs to direct the Th1-type immune response, which helps to further activate effector CD8^+^ T cells [6,7,8]. To date, available adjuvants belong to several classes of compounds: mineral salts, emulsions, and liposomes function in the storage of antigens (depot effect), while other classes of compounds, such as TLR agonists, cytokines, and plant-derived adjuvants, have pro-inflammatory effects. Presently, the utilization of aluminum salts as adjuvants in human vaccines is widespread [9] despite reported toxic effects, including allergenicity and neurotoxicity [10]. Other emulsions and liposomes can act as delivery vehicles that improve antigen stability and prevent degradation while also exerting immunomodulatory effects by optimizing antigen processing, inducing local inflammation, and accelerating APC recruitment and phagocytosis antigen uptake [11,12]. Some adjuvants, such as TLR agonists, have increasingly been studied as potential adjuvants for anticancer vaccines because of their crucial roles in the regulation of innate and adaptive immunity [13]. However, these adjuvants cannot elicit sufficient immune responses against antigens due to their predominant induction of Th2-type responses, while Th1-type immunity is essential for combating most intracellular pathogens and cancers [14,15]. Therefore, plant-derived adjuvants were focused on because they are available and can be produced easily and economically.

Recent studies have reported that major plant-derived adjuvants, such as saponins, polysaccharides, flavonoids, and plant virus-like particles, have become the focus of cancer vaccine adjuvants (Figure 1). It is well known that polysaccharides have pathogen-associated molecular patterns and could be bonded to pathogen-recognition receptors on the surface of APCs, which in turn modulate adaptive immunity [16]. In addition, natural polymers such as chitosan were used in cancer vaccines because they tend to be viscous, biodegradable, and non-toxic to humans [17]. Importantly, the inflammatory properties of chitosan can promote the polarization of immune cells toward either an anti-inflammatory or pro-inflammatory phenotype, making it an interesting candidate for cancer therapy [18,19]. In many studies, chitosan adjuvant vaccines increased the recruitment and number of cells associated with pro-inflammatory effects and reduced the number of anti-inflammatory cells, ultimately achieving the effects of reducing tumor weight, reducing metastasis, reversing immunosuppression, and improving survival in vivo [20,21]. Another well-known saponin adjuvant, QS-21, possesses remarkable adjuvant activity in cancer vaccines [22]. Clinical trials with ten conjugate saponin adjuvant vaccines in patients with either biochemically relapsed prostate cancer or castration-resistant metastatic prostate cancer following primary treatment demonstrated that saponin adjuvants can induce the production of numerous specific IgM and IgG antibodies [23]. The unique capacity of saponins to induce potent cellular immune responses, including those mediated by CD4^+^ helper T cells and CD8^+^ cytotoxic T lymphocytes, in addition to humoral immunity makes them good candidates for use as adjuvants in cancer vaccines [24].

However, these adjuvants may cause many side effects, including tenderness, mild redness, rare systemic symptom episodes, and itching at the vaccination site. For example, free antigens and adjuvants are cleared easily and rapidly, and the efficiency of targeted delivery to lymph nodes remains limited, so the released antigen alone is insufficient to achieve full activation of the anti-tumor immune response, which requires activation of APCs and the incorporation of immunomodulators. Also, some adjuvants have poor efficacy of drug delivery and uptake. Therefore, their production processes need to be improved by advanced nanotechnology. We found that there are six delivery nanoforms of these adjuvants used in cancer vaccine adjuvants: nanoemulsions, nanoparticles, nanoliposomes, nanosheets, nanogels, and nanomicelles (Figure 2). This review aimed to investigate the immunomodulatory properties of plant-derived vaccine adjuvants and their nanoadjuvants in various delivery systems to explore their applications in cancer-related vaccine research and to describe their pharmacological effects in cancer therapy.

## 2. Application of Plant Nanoadjuvants in Cancer Vaccines

### 2.1. Polysaccharide Adjuvant and Its Nanoadjuvants

Polysaccharides, which are polymerized glycans produced via plant cell metabolism, have been confirmed to have a strong anticancer effect and to relieve side effects [25]. Many polysaccharides, such as β-glucans, acidic polysaccharides from *Sarcandra glabra* (p-SGP) and *Portulaca oleracea* L. (POL-P3b), mannans, and polysaccharides from Astragalus (APS) and *Angelica sinensis*, have been used as adjuvants in cancer vaccines. These adjuvants can activate immune cells, facilitate the secretion of cytokines, and enhance protein stability. The anti-tumor effects of these adjuvants in cancer vaccines are exerted through the following process: (1) increasing antibody levels, (2) promoting secretion, (3) activating immune cells, and (4) enhancing antigen stability (Figure 3). For example, in Cheung’s study, the addition of β-glucan adjuvant to the GD2/GD3 vaccine induced a strong antibody response and improved the survival rate of mice. IgM and IgG1 titers rose at 6 weeks after immunization, peaking within a few months [26]. The clinical trial (phase II) found that oral administration of β-glucan during the first 5 weeks of prevaccination resulted in higher anti-GD2 IgG1 antibody responses (1.80; 90% CI, 0.12–3.39; *p* = 0.08; planned type I error, 0.10). Titers ≥ 230 ng/mL were associated with favorable progression-free survival at week 8. Moreover, this titer was significantly associated with the single nucleotide polymorphism rs3901533 of β-glucan receptor dectin-1 [27]. Yuba et al. developed a 100–200 nm diameter liposomal vaccine with pH-responsive polysaccharides. The binding ability of the pH-responsive modified liposomes to DC cells was 5-fold higher than the unmodified liposomes. Furthermore, by improving intratumoral immunosuppression and inducing cellular immunity, these liposomes are administered even at small doses of antigen [28]. p-SGP is regarded as another promising adjuvant for cancer vaccines. p-SGP significantly augmented the anti-tumor immunity of various cancer vaccines, which led to noticeable inhibition of tumor growth and metastasis in tumor-bearing mice. For example, after 28 days, the PD-L1-NitraTh vaccine administered without an adjuvant inhibited tumor growth with an inhibition rate of 40.17 ± 4.72%, while the PD-L1-NitraTh/p-SGP group had an inhibition rate of 68.64 ± 3.98%, which suggested that this adjuvant could greatly improve the anti-tumor effect [29]. Moreover, POL-P3b played a remarkable role in breast cancer treatment when it was combined with a dendritic vaccine as a novel adjuvant because cancer growth decreased by more than 50% [30]. Zhou et al. found that the IgG and IgG1 antibody titers of OVA+APS were higher than those of the OVA and OVA+CpG vaccine groups 8 weeks after immunization. It can promote DC maturation and improve the Th1-polarized immune response, and the TLR4/MyD88/NF-κB signaling pathway is involved in this immunomodulatory effect [29,31]. Furthermore, a vaccine administered with a mannan adjuvant could significantly enhance lymph node drainage and promote antigen capture by the dendritic cells [32]. Mixed polysaccharide adjuvants were also confirmed to be promising vaccine adjuvants. For example, an adjuvant with mixed polysaccharide fractions (*w*/*w*, 1:1, *Astragalus membranous* and *Codonopsis pilosulae*) was confirmed to not only stimulate increased dendritic cell expression of CD80/CD86 but also increase CD8^+^ T-cell accumulation in established tumors in the lungs, which may have been associated with increased anti-metastatic activity in vivo [33]. These studies suggested that many polysaccharides have been shown to improve APC activation, DC maturation, and the Th1-polarized immune response upon administration and ultimately result in robust tumor vaccine effects. Therefore, polysaccharides mainly activate adaptive immune responses both through IgG- or IgM-mediated humoral immunity and through CD4^+^ T-helper cell (Th)- or CD8^+^ cytotoxic T lymphocyte (CTL)-mediated cellular immunity.

A recent study found that oligosaccharides can also be used as adjuvants for cancer vaccines. In Wang’s study, an oligosaccharide-conjugate cancer vaccine prepared from the glycan of Globo-H (GH) and globo-series glycosphingolipid (GSL) aided in antigen presentation and intracellular enzymatic processing. This GH glycan-induced antibody specifically recognizes GH, while the SSEA3 glycan induces antibody target SSEA3 and exhibits cross-reactivity with both GH and SSEA4 due to the presence of a common epitope with SSEA3 glycan. The processed glycopeptides are primarily presented by DCs through MHC class II molecules, with a lesser contribution from MHC class I molecules, thereby triggering immune responses [34,35]. Data have shown that the inclusion of polysaccharide adjuvants with DC-based cancer vaccines can efficiently activate immune function by enhancing the production of cytokines that are involved in specific immune cell recruitment and differentiation processes [33]. Moreover, such adjuvants can stimulate the maturation of DCs, allowing them to present endogenous cancer antigens to immature T cells and initiate an immune response against tumor growth. For example, chitosan, the most widely studied polysaccharide adjuvant, promotes antigen presentation and immune responses [36]. In particular, this adjuvant can stimulate specific cellular immunity because it induces type I IFN production through the cGAS-STING pathway to activate DCs to induce a specific Th1 cell-related cellular immune response [37]. Meanwhile, the addition of glucomannan resulted in a decrease in the response of regulatory T cells by reducing the expression of the TGF-β and Foxp3 genes within the tumor microenvironment. These results demonstrate that using 2 mg of glucomannan as an adjuvant significantly enhanced the CTL response compared to groups treated with cell lysate + GM4, lysate vaccine, and PBS. As an adjuvant, 2 mg and 4 mg doses of glucomannan exhibited tumor suppression rates of 41.53% and 52.10%, respectively, when compared to the PBS group [38]. In addition, *Antrodia cinnamomea* polysaccharides can increase the production of pro-inflammatory factors and promote dendritic cell maturation through activation of TLR2/TLR4 and NF-κB/MAPK signaling pathways, leading to antigen-specific T-cell activation and differentiation into Th1 cells [39]. In Tu’s study, vaccines incorporating polysaccharide adjuvants from Crocus sativus petals immuno-edited the tumor microenvironment by promoting recruitment of immune cells, such as NK cells and CD8^+^T cells, while reducing immune suppressor cells, including MDSCs, TAMs, and Tregs along with their regulators; furthermore, it induced a shift from an M2-TAMs phenotype towards an M1 phenotype within tumor tissues. PCSPB (isolated and purified from crocus petals) significantly inhibited the growth of transplantable S180 sarcomas in mice in a dose-dependent manner, with inhibition rates of 36.9%, 42.5%, and 54.3% at doses of 2.5, 5.0, and 10 mg/kg, respectively [40]. Taken together, these findings show that polysaccharide adjuvants can be used in cancer vaccines to enhance immune responses. Many studies have reported that nanoadjuvant use in cancer vaccine therapy could improve the efficacy of immunotherapy [41]. It not only can protect antigens or adjuvants from enzymatic degradation but can also enhance antigenicity by encapsulating or binding to payloads [42,43], and such approaches have been used for cancer vaccines [44]. A polymeric nano-vaccine containing a pathogen-like mannan-decorated particle adjuvant could stimulate a robust immune response in a murine cancer graft model, yielding a cure rate of 50% with excellent anticancer effects [32]. Li et al. reported that dextran-derived nanoadjuvants can control the OVA antigen release process and enhance immunological activation [45]. Nanoliposomes with polysaccharide adjuvants, such as fucose, mannose, and glucan, activate endocytosis in APCs and can be conjugated with functional chemical groups to promote DNA interactions. For example, DOTAP nanoliposomes with a mannosylated adjuvant were generated to reach the lymph nodes by targeting mannose receptors on APCs and resulted in higher bone marrow-derived cell uptake of OVA antigens in vitro and in vivo [46]. Incorporation of a chitosan nanoadjuvant, modifications such as PEGylation, or the generation of polymer drugs can strengthen immune responses by modulating immunogenicity in the context of a cancer vaccine. For example, a cervical cancer vaccine with a chitosan adjuvant prepared with dendrimers conjugated to HPV E7/E6 peptides showed good anticancer immune effects after implantation in a murine model [47]. Miyamoto et al. created a potent vaccine adjuvant through DNA hybridization utilizing a CpG-ODN nanogel that was composed of CpG-ODN and β-glucan SPG. After hybridization between CpG oligonucleotides (CpG-ODNs), the size was shown to be between 10.1 and 150 nm, while the peak top value of Mw was 2 × 10^7^ g/mol. The cellular uptake studies showed that the fluorescence intensity of CL-CpG was almost 10-fold greater than those of f-SPG and CpG/f-SPG. The adjuvant activity of the CL-CpG nanogel demonstrated that it can increase secretion of IFN-γ and enhance CTL activity to attack tumor cells. In E.G7-OVA transplanted mice models, CL-CpG+OVA induced strong CTL responses and prevented tumor growth [48]. Gu et al. prepared a PLGA nanoparticle-based vaccine delivery system with immunomodulator angelica polysaccharide and OVA antigen to promote the immune response. The nanoparticles were spherical shaped, had an average size of 225.2 nm, were negatively charged, and had a high encapsulation efficiency of about 66.28%. Mice immunized with this vaccine exhibited significantly greater IgG levels at days 28 and 35, which revealed that it stimulated robust and sustained humoral immune responses. Additionally, it induced increased secretion of cytokines, such as IFN-γ, IL-6, and IL-10. These results confirmed that this nanoadjuvant vaccine possessed the potential to promote both Th1 and Th2 types of immune responses [49]. Yang et al. found that graphene oxide nanosheets (GO/PCP/OVA) with Poria cocos polysaccharides and PVA were around 120–150 nm in diameter. They induced about two- to three-fold upregulation of MHC II, CD86, and CD80 and also exhibited remarkably higher productions of IL-6 and IL-12, suggesting that the nanosheet induced BMDC maturation significantly [50]. Zhao et al. found that vaccine antigens which were encapsulated in ammonium chloride chitosan nanoparticles triggered the innate immune response after engaging the STING-cGAS pathway. When these nanoparticle vaccines were administered to RAW264.7 cells, the expression levels of IL-6, IL-12p40, and TNF-α increased in a dose-dependent manner [51]. Zhang et al. used the nanomaterials protamine sulfate and carboxymethyl β-glucan to form nanoparticles for CpG ODN as adjuvants, which were mixed with antigens of melanoma-derived tumor cell lysates and antigens for anti-tumor therapy. The concentration of INF-γ induced by the nanoadjuvants increased by 2.5-fold, indicating that this nanoadjuvant was able to trigger the cellular immune response and inhibit tumor growth [52]. Shufen et al. found that mannose-modified stearic acid-grafted chitosan (M-CS-SA) micelles (sizes: 100.27 ± 17.03 nm, zeta potential: 26.5 ± 1.31 mV) can absorb vaccine antigens, accelerate its entry into cytoplasm, and activate the intracellular cGAS-STING signaling pathway to enhance the anti-tumor effects. Furthermore, it also promotes the expression of CD86 and CD80 to facilitate the maturation of DCs [53]. These data suggested that polysaccharide adjuvants can improve the immune response upon administration and generate robust tumor vaccine effects.

### 2.2. Saponin Adjuvant and Its Nanoadjuvants

Saponins are mainly distributed in highland regions and in sea creatures such as starfish and sea cucumbers. Many vaccines based on saponin adjuvants, such as those from *Quillaja saponaria* and *Astragalus membranaceus*, have been found to expel phlegm, arrest coughing, and regulate immunity. There are data suggesting that saponin adjuvants could exert anti-tumor effects of cancer vaccines through the following process (Figure 4): (1) promoting antigen presentation and cellular uptake, (2) promoting secretion by inflammatory cells, and (3) promoting the activation of NRPL3. For example, the QS-21 saponin extracted from *Quillaja saponaria* and Astragalus saponin from *Astragalus membranaceus* are the most popular saponin adjuvants for cancer vaccines. The saponin QS-21, which is a triterpenoid glycoside, is a typical and widely studied adjuvant. Additionally, it was expected that by combining QS-21 with other adjuvants to form adjuvant systems, complementary adaptive immune effects could be obtained. For example, an adjuvant formulation containing monophosphoryl lipid A (MPLA) and QS-21 in liposomes (AS01) can act synergistically during the early IFN-γ response. Unfortunately, the phase III clinical results obtained for immunotherapy vaccines using QS-21/AS15 adjuvants in patients with NSCLC and melanoma have not been promising in terms of disease-free survival [54]. A study conducted on a lipid vaccine (lipo-saponins/bFGF) with Astragalus saponin adjuvants and a recombinant human basic fibroblast growth factor (bFGF) protein antigen reported significant anticancer effects in a CT26 lung cancer metastasis model. The authors found that the number of lung surface nodules was significantly decreased after treatment with lipo-saponins/bFGF. The results were significantly different from those in the control group (*p* < 0.05) [55]. Other relevant studies on Astragalus saponins have shown that they can indeed improve the cellular immune response. For example, the amount of IFN-γ released by Th1 cells in the saponin–aluminum hydroxide/bFGF group was substantially superior to those in the other groups. Furthermore, the lipo-saponins/bFGF group demonstrated superior CTL activity against MS1 (mouse vascular endothelial cells) (*p* < 0.01) [55,56]. ISCOMATRIX is another granular saponin adjuvant that contains cholesterol and phospholipids, and it can induce humoral and cellular immunity. In 2020, Jonathan S Cebon et al. studied the preliminary clinical effectiveness of the New York esophageal squamous cell carcinoma-1(NY-ESO-1) vaccine administered with the ISCOMATRIX adjuvant and reported strong antibody responses, although the titer of antibody declined over time. A prostate cancer vaccine with the ISCOMATRIX adjuvant, a Toll-like receptor 3 agonist, and an FMS-like tyrosine kinase ligand can achieve about 60% complete tumor regression. This immunological mechanism is attributed to the immune responses mediated by NK cells and CD4^+^T cells [57], and data suggest that saponin adjuvants are effective adjuvants for cancer vaccines [58]. These studies demonstrated that QS-21 in combination with other adjuvants has synergistic effects in promoting strong cellular immune responses, as confirmed by high amounts of IFN-γ production in animal models. Moreover, the known hemolytic properties of QS-21 can be avoided after co-administration with AS01 because of the presence of cholesterol in the liposome formulation [59]. Optimized semisynthetic natural QS-21 adjuvants were developed to overcome the heterogeneity and toxicities of the natural products. Pifferi found that QS-21-derived adjuvants that were covalently linked to a tumor-associated mucin-1-(glyco)peptide and a T-cell peptide could induce specific IgG antibodies to recognize the TA-MUC1 antigens on cancer cells [60]. It can stimulate both humoral and cellular immunity in combination with various vaccines and notably produces a stronger Th1-type immune response against intracellular pathogens than aluminum salts do. The mechanisms of QS-21 are summarized as follows: (1) The aldehyde group on the saponin interacts with T-cell receptors, such as CD2, to transmit costimulatory signals and accelerate APC presentation of cancer vaccine antigens to T cells, which can activate T cells and promote the secretion of Th1 cytokines. Therefore, this saponin has the capacity to promote the clearance of infected cells and plays a crucial role in the treatment of tumors. (2) Saponin activates the caspase 1-dependent NLRP3 inflammasome in mouse APCs, leading to secretion of IL-1β/IL-18. This process is associated with a Th1-type response.

In addition to QS-21, other saponin adjuvants are also used in cancer vaccines. For example, a natural triterpenoid saponin, Ginsenoside (Rg1), promoted secretion of cytokines, such as IL-6, TNF-α, IL-1β, and chemokines, such as IL-8 and IP-10, by peripheral blood mononuclear cells. In Huang’s study, Rg1 as a vaccine adjuvant with OVA induced much higher survival rates in C57BL/6 mice compared to mice immunized with OVA without adjuvants, since this adjuvant induces a potent anti-tumor immunity that polarized a Th1 immune response [61]. Ophiopogonin D (OPD), a vital natural steroidal glycoside from the root of *O. japonicas*, has been considered as an adjuvant candidate. However, its poor solubility and hemolytic toxicity hinder its use in human vaccination. Notably, saponins such as OPD and Tubeimuside-I, despite structural diversity, share functional synergy in cancer vaccines—enhancing cross-presentation and Th1/CTL responses—when optimized via nanotechnology to overcome solubility or toxicity barriers. In our studies, a self-nanoemulsifying adjuvant with OPD was prepared and contained the model antigen OVA to act as an anti-tumor vaccine. The antibody levels (including IgG, IgG1, IgG2a, and IgG2b) and cellular or humoral immune response (IFN-γ, IL-4, IL-1β, and IL-17A) both improved significantly. Importantly, this novel nanoadjuvant-based vaccine demonstrated preventive and treatment effects in lymphoma-bearing mice [62]. Another saponin adjuvant, tubeimuside-I (TBI), was found to be capable of enhancing an immune response and tumor suppression. Some research revealed that *F. nucleatum* correlated with the occurrence and progression of colorectal cancer (CRC). The nanoemulsion encapsulated with TBI (NTB) as a vaccine adjuvant could enhance the activation of *F. nucleatum* (*Fn*)-DCs in humoral and cellular immune responses. The NTB-*Fn*-DCs vaccine exhibited an excellent anti-tumor effect and improved long-term survival of CRC model mice [63]. These data suggested that saponin could be used in cancer vaccines.

Reportedly, saponin can stimulate the CTL response, which is necessary for inducing cancer vaccine immunity, so it could theoretically be a suitable adjuvant for cancer vaccines [55]. Previous studies have also shown that saponin adjuvants can induce antigen cross-presentation in DCs. For example, in 2016, Martijn H den Brok et al. proposed a mechanism by which saponin-based adjuvants could specifically induce antigen-presenting cells to produce MHC molecules in vitro and in vivo by recognizing GM-CSF on CD11b+ DCs. T-cell immunity can be enhanced by promoting MHC cross-presentation through intracellular liposomes [64]. Although there remain challenges ahead for the use of saponins as adjuvants, including their insolubility, hemolytic toxicity, and cost, these challenges must be overcome to enable their utilization in clinical applications, as saponin nanoadjuvants have shown explicit immunomodulatory effects and favorable efficacy in some preclinical studies.

### 2.3. Flavonoid Adjuvant and Its Nanoadjuvants

Flavonoids are widely found in plants, such as fruits, vegetables, grains, and trees (in the bark), and they have important biological activities, such as anti-tumor effects and preventive effects against many common cancers. Currently, flavonoids such as luteolin, chrysin, procyanidin, and hesperetin have emerged as promising adjuvants in cancer vaccines. Our findings demonstrate that these flavonoid adjuvants suppress tumor progression via mechanisms outlined in Figure 5: (1) enhancing CTL effects, (2) enhancing Th1 cell immune responses, (3) stimulating cytotoxic T cells, and (4) promoting APC antigen activation and presentation. For example, a vaccine with luteolin as an adjuvant had clear preventive and inhibitory effects on melanoma. In a mouse model, inactivated B16F10 cells were given a vaccine with luteolin as an adjuvant, and the authors found that luteolin could increase antigen presentation through the PI3K-Akt pathway, increase the expression of IL-4 cytokines to activate Th2 cells and IFN-γ to activate a Th1-type immune response, decrease regulatory T-cell numbers, and ultimately enhance CTL responses to increase tumor cell killing. It was confirmed in vivo that cancer cells grew more slowly in the group given the vaccine with the luteolin adjuvant than in the group given the vaccine alone, and those given the adjuvant had smaller final tumor masses and significantly improved 30-day survival rates [65]. Chrysin is another flavonoid that acts as a vaccine adjuvant. In a B16F10 melanoma cell model, the vaccine antigen mixed with chrysin activated APCs and enhanced Th1 cell effects by activating the IL2-STAT4 signaling pathway, thereby enhancing CTL anti-tumor responses. In comparison to the control group, the tumor size in the group that received the vaccine plus the adjuvant was significantly smaller and the survival rate was notably greater [66]. Another study showed that B16F10 cancer antigen plus a procyanidin adjuvant was also effective for inhibiting tumor growth and improving survival. The procyanidin adjuvant could similarly enhance the CD8^+^ T-cell-mediated immune response in vivo and serve as a promising adjuvant [67]. Hesperetin, a dihydroflavone, has been reported to have anticancer effects. Studies have shown that by stimulating cytotoxic CD8^+^ T cells, a vaccine of inactivated B16F10 melanoma cells with hesperetin adjuvant can activate APCs and enhance CTL immune responses. Compared to the B16F10 tumor vaccine alone, the vaccine combined with hesperetin effectively hindered tumor growth and boosted the survival rate of tumor-bearing mice [68]. The results of these studies demonstrate that flavonoid adjuvants can promote the activation of APCs and enhance CD8^+^ T-cell responses to tumor antigens in vivo. This evidence suggests that flavonoids are promising candidates for tumor vaccine development. Mohammed et al. conducted a study to investigate the total phenol and flavonoid contents in acacia honey from different altitudes against human cancer cell lines [69]. The data demonstrated that flavonoid adjuvants synergistically enhance immune activation and boost the therapeutic efficacy of cancer vaccines.

The specific mechanism involving flavonoids used as adjuvants in cancer vaccine applications has not been fully elucidated. Most studies have shown that these adjuvants are capable of enhancing the production of IL-12 through the PI3K-AKT pathway, leading to the activation of APCs, enhancement of Th1 cell function by activating the IL12-STAT4 signaling pathway, and, most importantly, induction of the cytotoxic effects of CTLs against cancer cells, which can inhibit the growth of cancer [65,67,68,70].

Zhu et al. demonstrated that flavonoid LW-213 could activate ER stress in a variety of tumor cells, including Hut-102, THP, and HeLa cells. Dynamic monitoring revealed CRT membrane translocation in the three cell lines treated with LW-213, and completion of CRT translocation from the ER membrane to the cell membrane occurred prior to apoptosis. Treatment of HUT102 cells with type M0 macrophages with LW-213 resulted in a transition to type M1 macrophages; it also reduced the number of type M2 macrophages [71]. Flavonoid adjuvants have been widely used in cancer vaccines, but because of their instability due to easy oxidation and their obvious toxic side effects, they still need to be modified and packaged.

Therefore, many nano-delivery forms for flavonoid adjuvants were reported. The nano-delivered quercetin adjuvant not only promoted apoptosis but also potentially activated innate immune pathways, such as the NLRP3 inflammasome or TLR4 signaling, which are critical for dendritic cell maturation and subsequent T-cell priming. Caspase-9 upregulation (9-fold vs. control) may reflect inflammasome involvement, while Bax/BCL-2 shifts (10-fold increase in pro-apoptotic Bax) could synergize with immunogenic cell death mechanisms to enhance antigen cross-presentation. For instance, in Alhakamy’s research, a nano-based delivery system was utilized to evaluate the cellular uptake and effectiveness of Quercetin (QRT) against lung cancer cells derived from human breast cancer MCF-7 cells. This system incorporated QRT flavonoids, peptides from scorpion venom (SV), and Phospholipon 90H (PL). The particle size of the system was 116.9 nm and the zeta potential was 31.5 mV. Compared with the control group, the level of Caspase-9, a key regulator of apoptosis, was significantly increased in the treatment group, and was about 9-fold higher than that in the control group. Bax and BCL-2 are proteins located in the cytoplasm that promote and inhibit apoptosis, respectively. Administration of QRT-PHM-SV resulted in a significant 10-fold increase in Bax protein expression compared with the control, while the expression of BCL-2 protein was only 0.2-fold lower than that of the control group [72]. These results indicate that flavonoid adjuvants play an important role in tumor vaccines.

### 2.4. Plant-Derived Virus-like Particle Adjuvant and Its Nanoadjuvants

Plant viral particles (PVPs), which have similar structures to VLPs, can work as vaccine conjugates to induce immune responses against murine B-cell malignancy when coupled to a weak idiotypic tumor antigen [73]. These adjuvants increase immune responses and have valuable features that make them appropriate for targeted drug delivery [74] through the following process (Figure 6): (1) T-cell activation, (2) innate immune response, (3) adaptive immune response, (4) TIN neutrophil growth, and (5) APC cell presentation. Plant-derived virus-like particles have the protein structures of plant viruses but lack genomic nucleic acids and are noninfectious [75]. The repetitive nature of plant virus-like particle structures is similar to that of PAMPPs, which makes them easily recognized by the immune system and good candidates for vaccine adjuvants [76]. So far, adjuvants in this class have mainly included cowpea mosaic virus (CPMV, an icosahedral structure), physalis mottled virus (PhMV), and papaya mosaic virus (PapMV).

Research has shown that self-assembling plant virus-like particles from CPMV administered via inhalation can not only inhibit B16F10 grafted lung melanoma but can also induce substantial anticancer effects against some poorly immunogenic cancers. The researchers found that in the B16F10 melanoma model, intratumorally administered formalin-inactivated CPMV suppressed tumor growth, prolonged mouse survival, and activated Toll-like receptor 7/8 signaling to reprogram the tumor microenvironment (TME) from immunosuppressive to immunostimulatory, thereby amplifying systemic anti-tumor immunity. Separately, inhaled CPMV inhibited lung melanoma via localized immune activation, as previously described [77]. It has also been reported that in situ vaccination with the CPMV can convert the tumor microenvironment from an immunosuppressive state to an immune activation state, thus leading to powerful anti-tumor immunity in tumor animal models. In short, CPMV could act as an adjuvant when combined with a CD47-targeted vaccine, and it was confirmed to have synergistic effects, decrease cancer growth [78], and induce immune effects [79]. Another adjuvant, PapMVa, could significantly promote the release of chemokines and pro-inflammatory cytokines and increase the infiltration of immune cells at tumor sites, promoting a significant increase in cancer-specific CD8^+^ T-cell numbers but a notable decrease in the proportion of myeloid suppressor cells after intertumoral administration. Importantly, this adjuvant also induced synergistic anti-tumor therapeutic effects by improving dendritic cell-based vaccine responses and strengthening PD-1 blocking to enhance the immune response [80]. For example, cowpea chlorotic mottle virus (CCMV) self-assembled with CpG oligonucleotides can promote the activation of tumor-associated macrophages and induce robust immune responses [81]. Studies by Hu showed that the anti-HER 2 vaccine in combination with a PhMV adjuvant can be successfully prepared by conjugating a human epidermal growth factor receptor 2 (Her 2)-derived CH401 epitope to the external surface of the virus-like particle. This vaccine may induce the production of high antibody titers and can increase the secretion of IFN-γ by Th1 cells and IL-6 by Th2 cells to promote the activity of CD4^+^ and CD8^+^ T cells, which have desirable synergistic effects for combating tumor growth and prolonging survival after subcutaneous administration in BALB/c mice using a prime-boost immunization schedule. Data have suggested the effective role of this vaccine in vivo against Her-2-expressing cancer [82]. In brief, while plant-derived virus adjuvants are pathogenic in plants, they are not only quite safe in animals but also important for cancer vaccines because they can overcome the immunosuppressive tumor microenvironment and subsequently stimulate immunological effects to treat metastatic tumor diseases.

Plant virus-based nanotechnology programs are creating drug delivery systems increasingly used in cancer vaccines. Such nanoparticles from tobacco mosaic virus (TMV) and potato virus X (PVX), due to their high width-to-height particle size ratios, were ideal platforms for drug delivery and the preparation of vaccine adjuvants [83]. For example, PVX nanoparticle adjuvants can act as carriers to deliver targeting peptides, and they can induce cancer cell apoptosis in breast cancer with high efficiency [84]. Another vaccine adjuvant used the cowpea mosaic virus (CPMV) and QβVLP to be recognized by toll-like receptors on immune cells within the lungs and sera; it increased the expression of IL-12 and interferon γ, promoting anti-tumor function, while it reduced the levels of IL-10 and transforming growth factor β, which have immunosuppressive effects. The effectiveness of this vaccine in preventing tumor metastasis was further confirmed in multiple tumor mouse models [85]. This is because the neutrophils in the lung tumor microenvironment are activated by the empty CPMV-like nanoparticle adjuvant, inducing an anticancer immune response. This adjuvant also exhibited anticancer effects in other tumor models, such as ovarian, colon, and breast cancer models [76]. It is reported that using the self-assembled virus-like nanoparticle adjuvant vaccine with cowpea mosaic virus (CPMV) can not only induce the effective immune response of B16V10 with poor immunogenicity but can also inhibit the growth of B16F10 lung melanoma tumors thorough activating neutrophils in the tumor microenvironment. These data suggested that this adjuvant vaccine had obvious therapeutic effects and systemic anti-tumor immune effects in ovarian, colon, and breast tumors [86]. It was also reported that CPMV conjugated with anti-PD-1 peptide (size about 30 nm) could improve anti-tumor effects in ovarian cancer mouse models [87]. Non-infectious CPMV nanoparticles were prepared with 1 mM formalin, 50 mm bPL, and 7.5 J CM^−2^ UV. It was found that UV-inactivated CPMV could maintain its structure, chemical reactivity, and, more importantly, its biological activity, while the formalin CPMV appeared to be more effective in in vivo studies of cutaneous melanoma [88]. It has been reported that both CPMV and eCPMV nanoparticles as novel adjuvants can overcome the immunosuppressive state and promote tumor shrinkage in ovarian cancer [89]. Also, CPMV could induce the release of pro-inflammatory cytokine IL-6, enhance the antigen-processing capacity of APCs, and initiate the activation of tumor-specific cytotoxic T cells in the tumor microenvironment. The mechanism of their immunomodulatory behavior is the transcription of single-stranded RNA present in the CPMV, which activates TLR7/8. TLR7 can drive the production of type I interferons, which are essential for the anti-tumor efficacy of RNA vaccines [89]. These data suggest that plant-derived virus-like particle adjuvants and their nanoadjuvants can improve the anti-tumor efficacy of cancer vaccines.

## 3. Limitations and Future Prospects

Plant-derived adjuvants and their nanoadjuvants can deliver targeted cancer therapeutic peptides that recognize receptors on DCs, stimulate the maturation of APCs through the PI3K-AKT pathway, and induce T-cell activation and differentiation through the TLR2/TLR4, NF-κB/MAPK, cGAS-STING, and IL-12-STAT4 signaling pathways, ultimately increasing the activity of CTLs to inhibit cancer growth (Figure 7). Cancer vaccines have shown promise in eliminating tumor cells and preventing metastasis by inducing antigen-specific T-cellular immunity. However, their effectiveness is limited by the presence of immunosuppressive cells, including M2 macrophages, and immunosuppressive cytokines, such as IL-10 and IL-13, which create an unfavorable cancer microenvironment [90]. However, the use of proper cancer-associated antigens, novel plant-derived immunostimulants, and drug delivery systems for cancer vaccines has addressed these problems by inducing a powerful immune response [91]. For example, trichosanthin (TCS), which is a kind of plant-derived protein drug, was also found to have effects on antigen presentation and can work as a vaccine adjuvant to strengthen anti-tumor immunity [92]. Guihua Chen evaluated a genetically engineered cancer vaccine in which the legumain domain acted as an antigen and an adjuvant of TCS and then further modified it with mannose. This mannosylated cancer vaccine could target dendritic cells and enhance antigen presentation via mannose receptor-mediated endocytosis. Furthermore, this “two-in-one” vaccine was encapsulated into a hydrogel, which promoted its sustained release characteristics. After being implanted into the postoperative site, this vaccine could play a long-term role in inhibiting the recurrence and metastasis of breast cancer [32]. In another study, the novel saponins tubeimuside-I (TBI) and ophiopogonin D that were added to a vaccine as adjuvants could also enhance the immune response [63,93] and inhibit tumor growth [94]. To reduce the hemolytic toxicity of TBI, it was encapsulated into a nanoemulsion with a DC vaccine (NTB-Fn-DCs). The results indicated that NTB-Fn-DCs markedly increased the activation of cytotoxic CD8^+^T lymphocytes and the long-term survival of a colorectal cancer mouse model [63]. Many plant-derived adjuvants have been studied worldwide, but further study is needed because they have not achieved optimal outcomes. For that reason, the use of these adjuvants with cancer vaccines still requires considerable research to ensure sustained and strengthened immune responses and maximization of their therapeutic benefits in the clinic. Currently, plant virus-based nanoparticle adjuvants such as TMV, CPMV, and PVX are still in small-scale production to optimize their applications in cancer vaccines. Ultimately, although further research and development are needed, plant-derived adjuvants are promising adjuvant candidates for cancer vaccines. 

## 4. Conclusions

In summary, the use of various plant-derived adjuvants such as saponins, polysaccharides, flavonoids, and plant virus-like particles has been applied in cancer vaccines because of their natural origins, low toxicities, and absence of harmful residues. At the same time, there are six delivery nanoforms of these adjuvants, including nanoemulsions, nanoparticles, nanoliposomes, nanosheets, nanogels, and nanomicelles, that are used for cancer vaccine adjuvants. These nanoadjuvant cancer vaccines not only promoted apoptosis but also activated immune pathways, which are critical for dendritic cell maturation and subsequent T-cell priming. Importantly, they can deliver targeted cancer therapeutic peptides that recognize receptors on DCs, stimulate the maturation of APCs through the PI3K-AKT pathway, and induce T-cell activation and differentiation through the TLR2/TLR4, NF-κB/MAPK, cGAS-STING, and IL-12-STAT4 signaling pathways, ultimately increasing the activity of CTLs to inhibit cancer growth. Therefore, plant-derived nanoadjuvants could be widely used in cancer vaccines.

## Figures and Tables

**Figure 1 vaccines-13-00378-f001:**
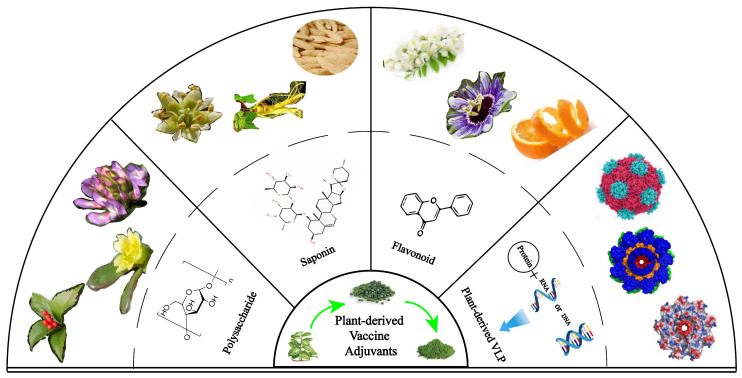
Diversity and applications of plant-derived vaccine adjuvants.

**Figure 2 vaccines-13-00378-f002:**
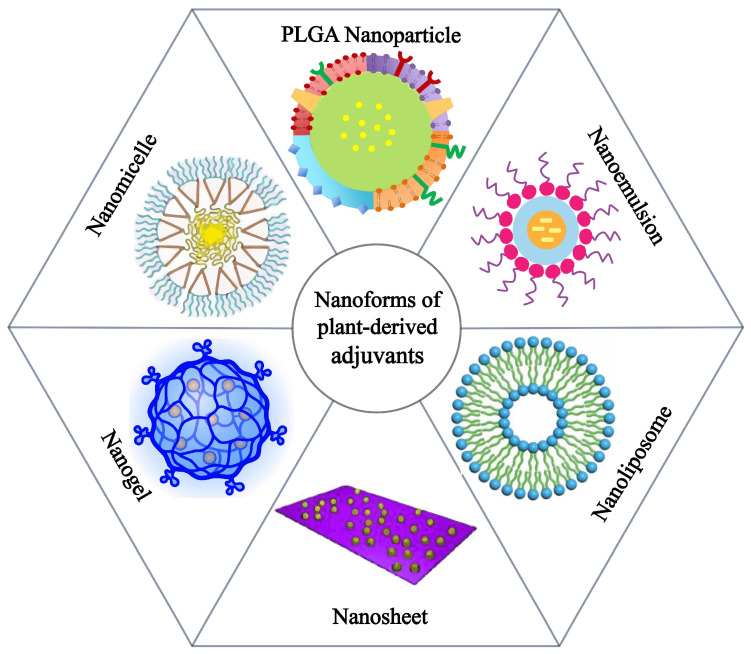
Various nanoforms of plant-derived vaccine adjuvants in cancer vaccines.

**Figure 3 vaccines-13-00378-f003:**
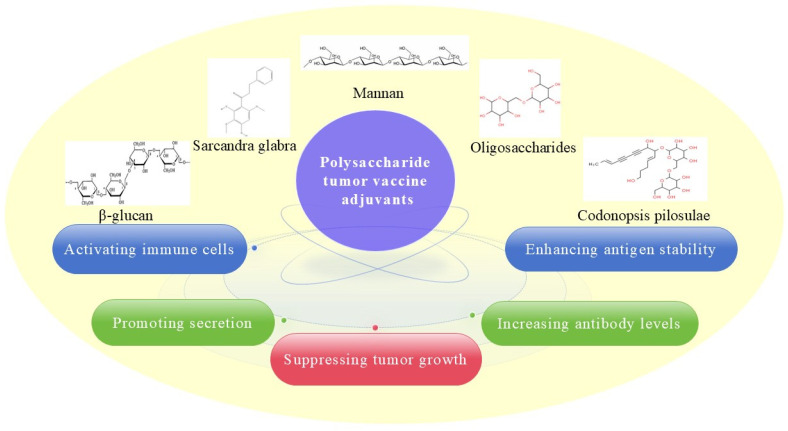
Anti-tumor process of polysaccharide adjuvants in cancer vaccines.

**Figure 4 vaccines-13-00378-f004:**
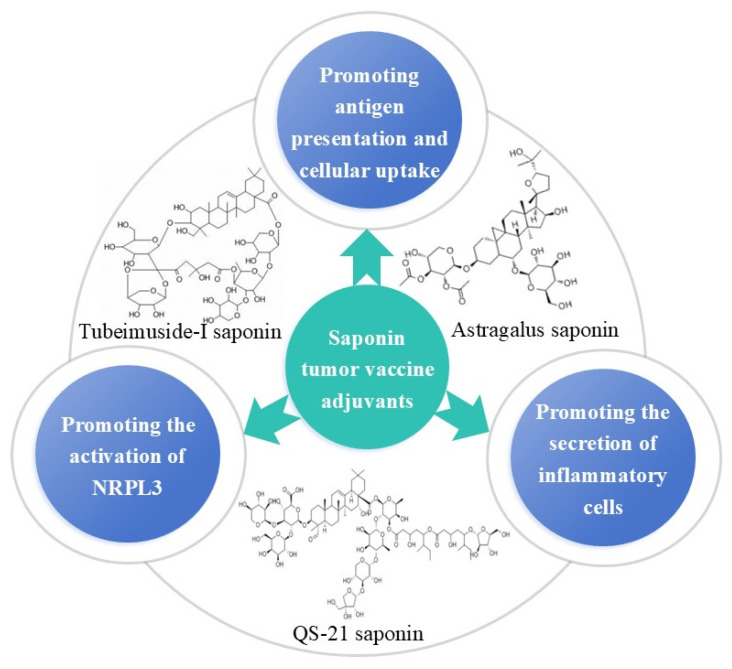
Anti-tumor effects of saponin adjuvants in cancer vaccines.

**Figure 5 vaccines-13-00378-f005:**
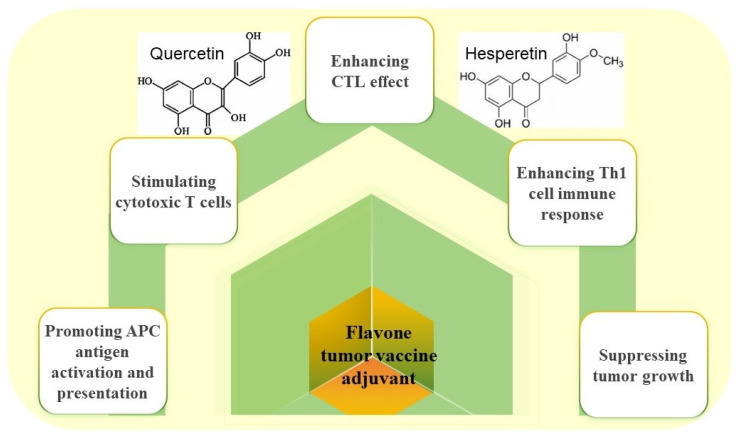
Anti-tumor effects of flavone adjuvants in cancer vaccines.

**Figure 6 vaccines-13-00378-f006:**
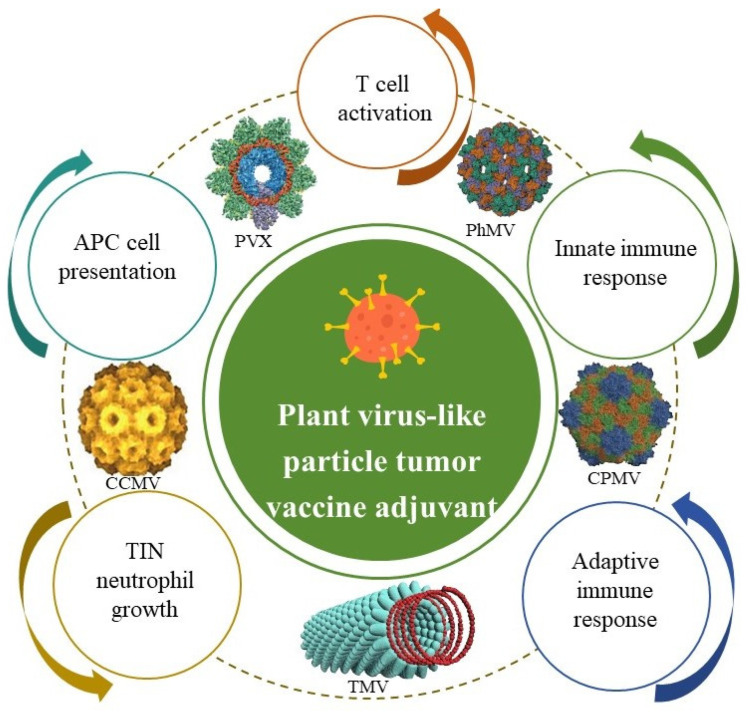
Anti-tumor effects of plant virus-like particle adjuvants in cancer vaccines.

**Figure 7 vaccines-13-00378-f007:**
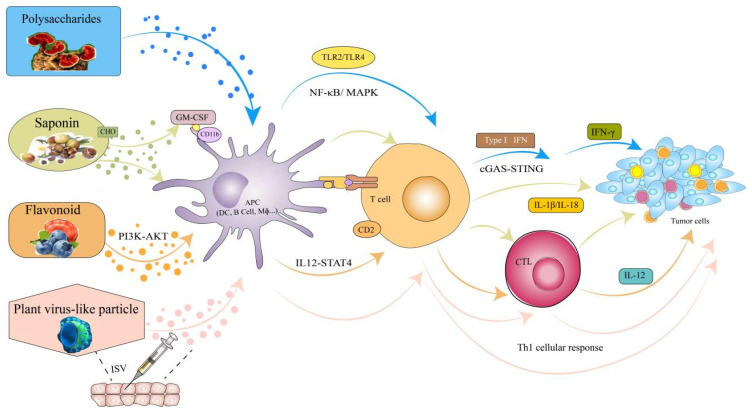
The improving anti-tumor mechanism of plant-derived adjuvants and their nanoadjuvants in combination with cancer vaccines.

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
