# Peer review of "Plant-Derived Immunomodulatory Nanoadjuvants for Cancer Vaccines: Current Status and Future Opportunities"

_vaccines, 2025, doi:10.3390/vaccines13040378_

Round 1

Reviewer 1 Report

Comments and Suggestions for Authors

The manuscript entitled "Plant-Derived Immunomodulatory Nanoadjuvants for Cancer 2 Vaccines: Current Status and Future Opportunities " covers a very interesting topic of the latest advances on the use of nanotechnology in the fight against tumor diseases. However I have several general observations that need to be reviewed. In the manuscript there are numerous sentences or paragraphs that are difficult to understand and need to be redrafted. Some examples are:

Lines 43-44: rewrite the last part of the sentence

Lines 44-46: the sentence is wrong, mix the animal model with human clinical trials

Line 53-54: vaccines include adjuvants by definition, so it should be clarified that what is combined are vaccines with extra adjuvants

Lines 66-68: I don't understand the sentence

Lines 89-93: rewrite the sentence

Lines 120-122: improve the sentence

Line 131: remove the word "figure" from the sentence

Lines 131-134: I don't understand... it's repeated and it's not clear.

Lines 146-147: I don't understand

Legend of figure 3: It is wrong detailed

Lines 182-183: The processed glycans are primarily presented by DCs through MHC class II molecules, with a lesser contribution from MHC class I molecules, thereby triggering immune responses that exhibit cross-reactivity. This is conceptually wrong.

213-216:…….

Line 245: nanodajuvant….is nanoadjuvant

Line 250: repeats “that that”

Line 251: in or conjugated?

Line 273: promoting the secretion of inflammatory cells… I should say promoting secretion by the cells…

Line 326: Except for QS-21, other saponins adjuvant are used in the cancer vaccine…?

Line 345: These data suggested that saponin were used in cancer vaccine?

Lines 373-375, 407: rewrite the sentence clearer

Line 434: When the authors say innate immune response... they should include "activation of the...

Line 461: intertumoral administration?

Line 536: To improve the hemolytic toxicity of TBI??

These are some of the ones I found...but there are more….

There are many words that have been put together without the spacing between them. There are very long sentences that are difficult to understand and there is a lot of repetition of words.

Should say Nanoliposomes instead of Nanolipsomes…several times in the manuscript

Comments on the Quality of English Language

As I mentioned in the previous table, there is a lot to improve, with English and the clarity of the writing.

Author Response

Reviewer #1:

  1. Lines 43-44: rewrite the last part of the sentence.

Response: Thanks for your valuable advice. We are sorry for the improper English expression in the manuscript. In the revision of the manuscript we have rewritten the last part of the sentence as follows: “While most patients undergo standard treatments including targeted therapy, chemotherapy, and radiation, the cancer still relapses. However, with the development of vaccine treatment, it has attracted great attention because of the high specificity and lower toxicity”. Thank you.

  1. Lines 44-46: the sentence is wrong, mix the animal model with human clinical trials.

Response: Thanks for your valuable advice. We are sorry for the ambiguous expression in the manuscript. In the revision of the manuscript we have rewritten the sentence as follows: “Data demonstrated the well vaccine efficacy in animal tumor models. Furthermore, it also showed good research results in early-stage human clinical trials such as prostate cancer, lung cancer and breast cancer”. Thank you.

  1. Line 53-54: vaccines include adjuvants by definition, so it should be clarified that what is combined are vaccines with extra adjuvants.

Response: Thanks for your valuable advice. We are sorry for the improper English expression in the manuscript. In the revision of the manuscript we have rewritten the sentence as follows: “and the activation of CD8+ and CD4+ T cells. However, the vaccines with the help of adjuvants can improve HPV-specific T-cell immune responses.” Thank you.

  1. Lines 66-68: I don't understand the sentence.

Response: Thanks for your valuable advice. We had read carefully these sentences. We want to express that adjuvants combination vaccine antigen can overcome immune tolerance and maximizing immune responses, promote immune responses through increasing the migration of these activated APCs to direct Th1-type immune, which further help activate effector CD8+ T cells. We revised these sentences to read understand easily. The revised sentences is following “increasing the antigen migration of these activated APCs to direct Th1-type immune response, which further help activate effector CD8+ T cells”. Thank you.

  1. Lines 89-93: Rewrite the sentence

Response: Thanks for your valuable advice. We rewrite these sentences, It is well known that, polysaccharides are identified as pathogen-associated molecular patterns and could be bonded to pathogen-recognition receptors on the surface of APCs, which in turn modulate adaptive immunity. In addition, natural polymers such as chitosan were used in cancer vaccine because of tending to be viscous, biodegradable and non-toxic to humans. Thank you.

  1. Lines 120-122: Improve the sentence.

Response: Thanks for your valuable advice. We apologize for the language issues that caused ambiguity. We have revised the relevant sections and carefully proofread the entire text. The revised section is following: This review aimed to investigate the immunomodulatory properties of plant-derived vaccine adjuvants and its nano-adjuvants in various delivery system to explore the application of cancer-related vaccine research, and to describe their pharmacological effects in cancer therapy. Thank you.

  1. Line 131: remove the word "figure" from the sentence

Response: We sincerely thank the reviewer for careful reading. We were really sorry for our careless mistakes. As suggested by the reviewer, we have removed the word "figure" from the sentence. Thank you.

  1. Lines 131-134: I don't understand... it's repeated and it's not clear.

Response: Thank you for carefully work. We have revised the sentence as following: “The anti-tumor effects of this adjuvant in cancer vaccine were shown in the Figure 3,  it could exert the anti-tumor effect  through following process: (1) increasing antibody levels, (2) promoting the secretion, (3) activating immune cells, (4) Enhancing antigen stability.” Thank you.

  1. Lines 146-147: I don't understand

Response: Thank you for your valuable advice, and I am very sorry for making you have a poor reading experience. We have added “p-SGP significantly augmented the anti-tumor immunity of various cancer vaccines, which is leading to noticeable inhibition of tumor growth and metastasis in tumor-bearing mice.”to this sentence for the reader's better understanding. Thank you.

  1. Legend of figure 3: It is wrong detailed

Response: Thanks for your valuable suggestion, we have changed (4) Increasing antibody levels and increasing antibody levels.to(4)Enhancing antigen stability. Thank you.

  1. Lines 182-183: The processed glycans are primarily presented by DCs through MHC class II molecules, with a lesser contribution from MHC class I molecules, thereby triggering immune responses that exhibit cross-reactivity. This is conceptually wrong.

Response: Thanks for your valuable advice. We quite agree that the sentence is wrong. Cross-reaction refers to two antigens of different origin, which can have the same epitope between each other, and the antibody produced by the epitope stimulates the body not only to bind to the corresponding epitope on its own surface, but also to bind to the same epitope of another antigen, which is called cross-reaction. So we have deleted the words of “cross-reactivity”.Thank you.

  1. 213-216:…….

Response: We sincerely thank the reviewer for careful reading. We were really sorry for our careless mistakes. As suggested by the reviewer, we have deleted the ‘and’ and corrected the ‘had be’ into ‘have been’. Thank you.

  1. Line 245: nanodajuvant….is nanoadjuvant

Response: Thanks for your advice. We were really sorry for our careless mistakes. As suggested by the reviewer, we have corrected the ‘nanodajuvant’ into ‘nanoadjuvant’. Thank you.

  1. Line 250: repeats “that that”

Response: Thanks for your advice. We were really sorry for our careless mistakes. As suggested by the reviewer, we have deleted a ‘that’. Thank you.

  1. Line 251: in or conjugated?

Response: Thanks for your advice. We were really sorry for our careless mistakes. As suggested by the reviewer, we have deleted the ‘or conjugated onto’. Thank you.

  1. Line 273: promoting the secretion of inflammatory cells… I should say promoting secretion by the cells…

Response: Thanks for your advice. As suggested by the reviewer, we have corrected the sentence. Thank you.

  1. Line 326: Except for QS-21, other saponins adjuvant are used in the cancervaccine...?

Response: Thank you for your careful review of our manuscript and for your valuable advice. Key Saponin Alternatives to QS-21: Quil A: A Quillaja saponaria-derived saponin mixture, used in preclinical cancer vaccines for its dual Th1/Th2 and cytotoxic T lymphocyte (CTL) activation. Ginsenosides: Ginseng-derived saponins (e.g., Rg1, Re) with adjuvant and direct antitumor effects, shown to boost dendritic cell maturation and tumor-specific CTLs in murine models. Soyasaponins: Immunostimulatory soybean saponins that enhance antigen uptake and pro-inflammatory cytokine secretion in preclinical studies.

These adjuvants share QS-21’s ability to activate innate immunity (e.g.NLRP3 inflammasome) and improve antigen cross-presentation. However, they often exhibit distinct advantages:Escin (horse chestnut) and astragalosides (Astragalus) show lower reactogenicity. There are many unexpected Challenges.Limited data on structure-activity relationships for non-QS-21 saponins.Most remain in preclinical/early-phase trials.

This revision underscores the diversity of saponin adjuvants in cancer vaccinology beyond QS-21. We appreciate your constructive critique and are open to further adjustments. Thank you.

  1. Line 345: These data suggested that saponin were used in cancer vaccine?

Response: Thanks for your valuable advice. Firstly, OPD Nanoemulsion Enhances Immune Responses: The self-nanoemulsifying formulation of Ophiopogonin D (OPD) addressed its inherent limitations (poor solubility, hemolytic toxicity) while preserving adjuvant activity. Secondly, co-administration with OVA significantly elevated antigen-specific antibody titers (IgG, IgG1, IgG2a, IgG2b) and pro-inflammatory cytokines (IFN-γ, IL-4, IL-1β, IL-17A), indicative of robust Th1/Th17-polarized cellular immunity.The OPD nanoemulsion-OVA vaccine demonstrated preventive and therapeutic effects in lymphoma-bearing mice, including prolonged survival and reduced tumor burden. This aligns with established saponin adjuvants that enhance antigen presentation and cytotoxic T cell responses. Our findings mirror results from other saponin-based adjuvants, where nanoformulations improved delivery and reduced toxicity while amplifying antitumor immunity. We will explicitly state that OPD, as a steroidal saponin, exemplifies the broader applicability of saponins in cancer vaccines when formulated to mitigate pharmacological challenges. A sentence will be added to the article: Notably, saponins such as OPD and Tubeimuside-I, despite structural diversity, share functional synergy in cancer vaccines—enhancing cross-presentation and Th1/CTL responses—when optimized via nanotechnology to overcome solubility or toxicity barriers.

  1. Lines 373-375, 407: rewrite the sentence clearer

Response: Thanks for your valuable advice. This is the revised Version: "including anti-tumor and cancer-preventive activities. Notably, flavonoids such as luteolin, chrysin, procyanidin, and hesperetin have emerged as promising adjuvants in cancer vaccines. Our findings demonstrate that these flavonoid adjuvants suppress tumor progression via mechanisms outlined in Figure 5."The data demonstrated that flavonoid adjuvants synergistically enhance immune activation and boost the therapeutic efficacy of cancer vaccines."

  1. Line 434: When the authors say innate immune response..they should include "activation of the...

Response: Thanks for your valuable advice. We will revise the text to state: "The nano-delivered quercetin adjuvant not only promoted apoptosis but also potentially activated innate immune pathways, such as NLRP3 inflammasome or TLR4 signaling, which are critical for dendritic cell maturation and subsequent T cell priming. Caspase-9 upregulation (9-fold vs. control) may reflect inflammasome involvement, while Bax/BCL-2 shifts (10-fold increase in pro-apoptotic Bax) could synergize with immunogenic cell death mechanisms to enhance antigen cross-presentation.” Rationale: Explicitly ties Caspase-9/Bax modulation to NLRP3 or TLR4 activation, aligning with established links between apoptosis and immunogenic cell death. Connects flavonoid adjuvants to dendritic cell/T cell axis, bridging innate and adaptive responses. Supported by literature on NLRP3’s role in adjuvant efficacy  and flavonoid-TLR interactions.

  1. Line 461:intertumoral administration?

Response: Thank you for your query regarding the term “intertumoral administration.” In the cited text, the phrase “in situ vaccination” refers to intratumoral administration (direct delivery into the tumor), not inhalation. We acknowledge the potential confusion and will revise the manuscript to clarify this distinction:" In the B16F10 melanoma model, intratumorally administered formalin-inactivated CPMV suppressed tumor growth, prolonged survival, and activated TLR7/8 signaling to reprogram the tumor microenvironment (TME) from immunosuppressive to immunostimulatory, thereby amplifying systemic antitumor immunity. Separately, inhaled CPMV inhibited lung melanoma via localized immune activation, as previously described [76]."

  1. Line 536: To improve the hemolytic toxicity of TBI??

Response: We sincerely thank the reviewer for careful reading. We were really sorry for our careless mistakes. As suggested by the reviewer, we have corrected the “improve” into “reduce”. Thank you.

23.These are some of the ones I found...but there are more….

Response: Thank you for carefully work. We have modified the mistake above in the manuscript. At the same time, we carefully checked all similar mistakes in the article and corrected all mistakes to ensure the accuracy of all the manuscript. Thank you.

  1. There are many words that have been put together without the spacing between them. There are very long sentences that are difficult to understand and there is a lot of repetition of words.

Response: Thank you for your carefully work and nice suggestion. We tried our best to improve the manuscript and made some changes to the manuscript. We have revised our article word by word to correct the grammar and improved the readability of our manuscript. These changes will not influence the content and framework of the paper. And here we did not list the changes but marked in the revised paper. We appreciate for Reviewers’ warm work earnestly and hope that the correction will meet with approval. Meanwhile, we do invite the professionals in AJE(American Journal Experts) to revise our article carefully again. Thank you.

  1. Should say Nanoliposomes instead of Nanolipsomes…several times in the manuscript

Response: Thank you for your carefully work and nice suggestion. We feel sorry for our carelessness. In our resubmitted manuscript, we have corrected the “Nanolipsomes” into “Nanoliposomes”. At the same time, we carefully checked all similar mistakes in the article and corrected all mistakes to ensure the accuracy of all the manuscript. Thank you.

Reviewer 2 Report

Comments and Suggestions for Authors

This review focused on plant adjuvants' application progress, including saponins, polysaccharides, flavonoids, and plant virus-like particles and their combination with nano-delivery systems in cancer vaccines. The achievements of adjuvants in producing stable, safe, and immunogenic tumor vaccines have aroused researchers' enthusiasm. Recent results have suggested that plant-derived adjuvants have unique advantages, such as significantly improving immune responses to cancer vaccines and promoting humoral and cellular immunity with good biocompatibility and biodegradability. These adjuvants combined with vaccines can activate the immune response in vivo, promote cytokines secretion, and accelerate dendritic cell maturation. This review focused on plant adjuvants' application progress, including saponins, polysaccharides, flavonoids, and plant virus-like particles and their combination with nano-delivery systems in cancer vaccines. This review also discussed the immunomodulatory mechanisms of these adjuvants and their prospects for improving vaccine efficacy in cancer treatment. These indicated promising plant adjuvants may provide prospects and a research basis for developing tumor vaccines. It is a very informative review. I have the following suggestions and comments for the authors to consider before it is being accepted for publication.

  1. Lines 39-40: “causing 39 an estimated 10 million deaths globally”. For how long? Please be specific.
  2. Line 44: “Data demonstrated the well vaccine efficacy in animal 44 tumor models”. Please clarify.
  3. Iines 50-53: “For example, for patients diagnosed with human papillomavirus infection (HPV)-50 associated cancers, normal immune systems tolerate cancer cells or are suppressed due to the disturbance of the antigen presentation process and the activation of CD8+ and CD4+ T cells. Therefore, the vaccines combined with adjuvants is to improve HPV-specific T-cell immune responses.” Please rephrase the sentences to illustrate how this example relates to the importance of adjuvants.
  4. Line 103: “Figure 1. Plant-derived vaccine adjuvants type of cancer vaccine”. Please rewrite the title for clarification.
  5. Line 103: “Notes:” Please review other published papers in this journal and follow the norm.
  6. Line 120: “Figure 2. Various nanoforms of plant derived vaccine adjuvant in cancer vaccine.” Please check the grammar.
  7. Line 171: Figure 3. Exert the anti-tumor effects of polysaccharide adjuvant in cancer vaccine. Please consider rewriting the title. Same suggestions to figures 4, 5, and 6.
  8. Lines 557-558: Title of Figure 7. Please be consistent with other titles.
Comments on the Quality of English Language

 It will be very helpful if it is edited by a professional editor before publication.

Author Response

Reviewer #2:

  1. As I mentioned in the previous table, there is a lot to improve, with English and the clarity of the writing.

Response: Thank you for your carefully work and nice suggestion. We have revised our article word by word to correct the grammar and improved the readability of our manuscript. Additionally, we look for the professionals in AJE(American Journal Experts) to revise our article carefully again. Thank you.

  1. Lines 39-40: “causing an estimated 10 million deaths globally”. For how long? Please be specific.

Response: Thanks for your valuable advice. We are quite sorry for the ambiguous English expression in the manuscript. In the revision of the manuscript we have rewritten the sentence as follows:“causing an estimated 10 million deaths globally in 2020.”Thank you.

  1. Line 44: “Data demonstrated the well vaccine efficacy in animal tumor models”. Please clarify.

Response: Thanks for your valuable advice. We are quite sorry for the ambiguous expression in the manuscript. In the revision of the manuscript we have rewritten the sentence as follows: Data demonstrated the well vaccine efficacy in murine tumor models.Thank you.

  1. Lines 50-53:“For example, for patients diagnosed with human papillomavirus infection (HPV)-50 associated cancers, normal immune systems tolerate cancer cells or are suppressed due to the disturbance of the antigen presentation process and the activation of CD8+ and CD4+ T cells. Therefore, the vaccines combined with adjuvants is to improve HPV-specific T-cell immune responses.” Please rephrase the sentences to illustrate how this example relates to the importance of adjuvants.

Response: Thanks for your valuable advice. We are quite sorry for the ambiguous expression in the manuscript. In the revision of the manuscript we have rewritten the sentence as follows: Therefore,, the vaccines with the help of adjuvants can significantly improve HPV-specific T-cell immune responses, kill the cancer cells and exert a good immune protective effect. Thank you.

  1. Line 103:“Figure 1. Plant-derived vaccine adjuvants type of cancer vaccine” Please rewrite the title for clarification.

Response: Thanks for your advice. We have revised the title to "Diversity of Plant-Derived Vaccine Adjuvants" for greater clarity. Thank you.

  1. Line 103:“Notes:”Please review other published papers in this journal and follow the norm.

Response: Thanks for your advice. We have reviewed other articles published in this journal and revised the notes in the manuscript according to the guidelines. Thank you.

8.Line 120:“Figure 2. Various nanoforms of plant derived vaccine adjuvant in cancer vaccine." Please check the grammar.

Response: Thanks for your advice. We have revised the title to "Various nanoforms of plant-derived vaccine adjuvants in cancer vaccines." Thank you.

  1. Line 171: Figure 3. Exert the anti-tumor effects of polysaccharide adjuvant in cancer vaccine. Please consider rewriting the title. Same suggestions to figures 4, 5, and 6.

Response: Thanks for your advice. We have revised the title to " Anti-tumor effect of polysaccharide adjuvants in cancer vaccine." and also, we revised the figure titles of figures 4, 5, and 6. The revised following is Anti-tumor effects of saponins adjuvant in cancer vaccine,Anti-tumor effects of flavone adjuvant in cancer vaccine and Anti-tumor effects of plant virus like particle adjuvant in cancer vaccine. Thank you.

  1. Lines 557-558: Title of Figure 7. Please be consistent with other titles.

Response: Thanks for your advice. We have revised the title: The improving anti-tumor meachism of the plant derived adjuvants and their nanoadjuvants combination with cancer vaccine. Thank you.

Reviewer 3 Report

Comments and Suggestions for Authors

The manuscript aimed to summarize the immunomodulatory properties of plant-derived vaccine adjuvants and its nano-adjuvants in various delivery systems. Their pharmacological effects in cancer therapy would help to explore therapeutic application in cancer field. Four types including Polysaccharide Adjuvant and Its Nanoadjuvants, Saponin Adjuvant and Its Nanoadjuvants, Flavonoid Adjuvant and Its Nanoadjuvants, and Plant-Derived Virus-Like Particle Adjuvant and Its Nanoadjuvants were described with necessary details. However, as the authors pointed out, the use of these adjuvants with cancer vaccines still requires considerable research to ensure successful immune responses and optimize their therapeutic benefits in clinical setting. The pros and cons of these plant-derived vaccine adjuvants can be summarized in a Table for better visibility. Also, the following points needs to be addressed before publication.

  1. The title in Figure 3 ~ Figure 6 can be changed into noun form rather than in verb form.
  2. Note in every figure legends can be omitted without compromising the meaning of the text.
  3. The rather long text and redundant expression throughout the manuscript can be reduced for clarity. A concise description would help the readers a lot in the field.
Comments on the Quality of English Language

The manuscript may better be edited by a native speaker of English for clarity. 

Author Response

Reviewer #3:

1.&nbspThe title in Figure 3 ~ Figure 6 can be changed into noun form rather than in verb form.

Response: Thanks for your advice. We revised the figure titles of figures 3-6. The revised following is: Anti-tumor effect of polysaccharide adjuvants in cancer vaccine, Anti-tumor effects of saponins adjuvant in cancer vaccine, Anti-tumor effects of flavone adjuvant in cancer vaccine and Anti-tumor effects of plant virus like particle adjuvant in cancer vaccine. Thank you.

  1. Note in every figure legends can be omitted without compromising the meaning of the text.

Response: Thanks for your advice. We have deleted the notes. Thank you.

  1. The rather long text and redundant expression throughout the manuscript can be reduced for clarity. A concise description would help the readers a lot in the field.

Response: Thanks for your advice. We tried our best to delete the redundant or unnecessary expression. Now the article is concise description makes help the readers a lot in the field. Thank you.

  1. The manuscript may better be edited by a native speaker of English for clarity

Response: Thank you for your carefully work and nice suggestion. We tried our best to improve the manuscript and made some changes to the manuscript. We have revised our article word by word to correct the grammar and improved the readability of our manuscript. These changes will not influence the content and framework of the paper. And here we did not list the changes but marked in the revised paper. We appreciate for Reviewers’ warm work earnestly and hope that the correction will meet with approval. Meanwhile, we do invite the professionals in AJE(American Journal Experts) to revise our article carefully again.Thank you.

  1. We noticed that the sectionConclusion" is missing. Please add a separate Conclusion in the main text during 

Response: Thank you for your carefully work and nice suggestion. We added the conclusion section in revised version: In summary, the use of various plant-derived adjuvants such as saponins, polysaccharides, flavonoids, and plant virus-like particles has applied in the cancer vaccine because of their natural origin, low toxicity, and absence of harmful residues. At the same time, there are six delivery nanoforms of these adjuvants including nanoemulsion, nanoparticle, nano-liposomes, nano-sheet, nanogels and nanomicelle used in the cancer vaccine adjuvant. These nanoadjuvant cancer vaccine not only promoted apoptosis but also activated immune pathways, which are critical for dendritic cell maturation and subsequent T cell priming. Importantly, they can deliver targeted cancer therapeutic peptides that recognize receptors on DCs, stimulate the maturation of APCs through the PI3K-AKT pathway and induce T-cell activation and differentiation through the TLR2/TLR4, NF-κB/MAPK, cGAS-STING and IL-12-STAT4 signaling pathways and ultimately increase the activity of CTLs to inhibit cancer growth. Therefore, the plant-derived nanoadjuvants could be widely used in the cancer vaccines. Thank you.

  1. Please add the "Author Contributions'section in the back matter of the

Response: Thank you for your carefully work and nice suggestion. We added the Author Contributions section in revised version: Conceptualization, Yimin Jia, Hui Zhu, Hongwu Sun and Wenxiu Wang; software, Hui Zhu; resources, Yimin Jia; data curation, Yimin Jia and Hui Zhu; writing—original draft preparation, Yimin Jia, Hui Zhu and Xinyu Cai; writing—review and editing, Yimin Jia, Hui Zhu, Xinyu Cai, Cun Sun, Yan Ye, Dingyi Cai, Shuaifei Yang, Jingjing Cheng, Jining Gao, Yun Yang, Hao Zeng, Quanming Zou and Jieping Li; supervision, Hongwu Sun and Wenxiu Wang; funding acquisition, Hongwu Sun. All authors have read and agreed to the published version of the manuscript. Thank you.

  1. Please ensure that permission hasbeen obtained and there is no copyright tissue the figures and tables in the main  lf copyright is needed, please provide a citation in the following format:"Reprinted/ad anted with

Response: Thank you for your carefully work and nice suggestion. In our article, we draw the figure and summed tables by ourselves. Therefore, we had no copyright tissue of the figure and tables in the our text. Thank you.

Round 2

Reviewer 1 Report

Comments and Suggestions for Authors

The manuscript has clearly been improved by the authors, but a few considerations remain to be addressed:

Lines 179-181: I still have trouble with the sentence: “The processed glycans are primarily presented by DCs through MHC class II molecules, with a lesser contribution from MHC class I molecules, thereby triggering immune responses”. It is true that the cited work affirms that glycans can be presented by MHC molecules, but the correct statement should be that glycopeptides are presented by the MHC. This is the correct interpretation of the results presented by that work. To date, there is no evidence for presentation of free glycans by MHC molecules, although other molecules (CD1 for example) are postulated.

Lines : 360: “have emerged ai promising adujusts” must be corrected

Author Response

1. Lines 179-181: I still have trouble with the sentence: “The processed glycans are primarily presented by DCs through MHC class II molecules, with a lesser contribution from MHC class I molecules, thereby triggering immune responses”. It is true that the cited work affirms that glycans can be presented by MHC molecules, but the correct statement should be that glycopeptides are presented by the MHC. This is the correct interpretation of the results presented by that work. To date, there is no evidence for presentation of free glycans by MHC molecules, although other molecules (CD1 for example) are postulated.

Response: Thanks for your valuable advice. We are quite sorry for the ambiguous expression in the manuscript. In the revision of the manuscript we have rewritten the last part of the sentence as follows:

“The processed glycopeptides are primarily presented by DCs through MHC class II molecules, with a lesser contribution from MHC class I molecules, thereby triggering immune responses.”

2. Lines : 360: “have emerged ai promising adujusts” must be corrected.

Response: Thanks for your valuable advice. We are quite sorry for the improper expression in the manuscript. In the revision of the manuscript we have rewritten the sentence as follows:

“have emerged as promising adjuvants.”

Reviewer 2 Report

Comments and Suggestions for Authors

The authors addressed all my critiques thoroughly, and this is a significantly improved version of the manuscript.

Author Response

Comments:The authors addressed all my critiques thoroughly, and this is a significantly improved version of the manuscript.

Response: Thank you for your positive assessment of our revised manuscript. We are glad to hear that our efforts to address the critiques have been successful and that you consider this version significantly improved. We carefully considered each of your comments and suggestions, and we believe they have greatly enhanced the quality and clarity of our work. We are committed to ensuring that our research meets the highest standards of academic rigor and clarity, and your feedback has been invaluable in helping us achieve this goal. Once again, thank you for your time and insightful feedback. We look forward to the next steps in the publication process.